# Cross-Sectional Analysis of Colombian University Students’ Perceptions of Mental Health during the COVID-19 Pandemic: Repercussions on Academic Achievement

**DOI:** 10.3390/healthcare11142024

**Published:** 2023-07-14

**Authors:** María Alejandra Camacho-Villa, Ingrid Johanna Díaz-Marín, Erika Tatiana Paredes Prada, Adrián De la Rosa, Gloria Isabel Niño-Cruz

**Affiliations:** 1Sport and Innovation Research Group (GICED), Laboratory of Applied Sciences of Sport, Unidades Tecnológicas de Santander (UTS), Bucaramanga 680006, Santander, Colombia; ingridd@correo.uts.edu.co (I.J.D.-M.); eparedes@correo.uts.edu.co (E.T.P.P.); adelarosa@correo.uts.edu.co (A.D.l.R.); 2Pain Study Group (GED), Physical Therapy School, Universidad Industrial de Santander, Bucaramanga 680002, Santander, Colombia; 3Harmony and Life Movement (MAV), Physical Therapy School, Universidad Industrial de Santander, Bucaramanga 680002, Santander, Colombia; glonicru@correo.uis.edu.co

**Keywords:** anxiety, depression, academic performance, COVID-19, university students

## Abstract

Background: During the COVID-19 pandemic, the increase in academic activities at home negatively impacted mental health, due to significant levels of stress, anxiety, and depression. We investigated the association of depression and anxiety with university students’ academic performance during the COVID-19 lockdown in Bucaramanga, Colombia. Materials and Methods: A cross-sectional study was conducted through an online survey during the lockdown, through the use of the Hospital Anxiety and Depression Scale (HADS) and the grade-point average. Results: 1090 females, out of 1957 students, with mean age 22.2 ± 5.3 years, participated in the study. The prevalence of low academic performance was higher in men (4.7% *p* = 0.014). As for mental health, 2.5% of the students were classified as “definite cases” of anxiety and 8.2% were diagnosed with depression. Women with a low academic performance had a greater percentage of being “definite cases” (3.8%) of anxiety as compared to men (1.1%). Regarding depression, in men, 12.2% of those with low academic performance were “definite cases” (6.9% *p* < 0.05); in females with low academic performance, 10.1% were “definite cases” of depression, according to the Poisson regression analysis. The probability of low performance was 100% higher for women identified as “doubtful cases” of depression (PR = 2.0; 95% CI: 1.10–5.18, *p* = 0.03). Conclusions: A positive association between the severity of anxiety/depression and lower grades, especially in women, was found. Mental health should be a special concern when considering university students, in order to improve their academic performance.

## 1. Introduction

The COVID-19 pandemic was declared by the World Health Organization (WHO) in March 2020. From that moment on, more than 650 million positive cases and almost 6.9 million deaths worldwide have been reported, from 11 January 2020 to 12 June 2023 [1]. The COVID-19 pandemic remains an issue and has considerably changed people’s daily lives. The Colombian Ministry of Health and Social Protection still reports new cases, demonstrating that the virus continues to affect the physical and mental conditions of citizens [2]. This situation continues to demand attention from researchers to better understand the dramatic impact of the COVID-19 lockdown on different population groups, such as university students.

In March 2020, Colombia declared a mandatory confinement, which was progressively extended, with new exceptions. Although the reopening of the country took place in September 2020, most of the economic, productive, and educational sectors modified their routines, through the implementation of work-from-home policies. This substantially increased the time dedicated to sedentary behaviors (especially increased screen time), as well as causing a notable decrease in the practice of physical activity related to transport and leisure activities [3,4].

In the education sector, university students were forced to adapt to new environmental, technological, and psychosocial conditions to receive their education virtually [5].

Numerous studies have been performed worldwide to assess the impact of COVID-19 on the mental health of university students [6,7,8,9,10,11,12]. For instance, in 2020, Wang X et al. [13] reported higher levels of depression (48.14%, *n* = 960) and anxiety (38.48%, *n* = 775) as a result of the COVID-19 lockdown in a sample of 2031 university students in the United States. Likewise, 1156 university students were surveyed in China during the same period of time. Researchers found symptoms of depression and anxiety in 48.1% and 57.6% of the participants, respectively [14]. An Italian study with 3533 university students reported a prevalence of 13.1% for anxiety and 32.3% for depression during the lockdown [15].

Similarly, a recent report by our research group, in a private Colombian University, showed a greater effect on the psychological well-being of 298 students, especially females, during the lockdown [16]. The evidence above indicates that, during the pandemic, these conditions enhanced and negatively impacted mental health in this population. It is possible that adverse situations such as social distancing, the unexpected length and severity of the outbreak, as well as familiar and economic problems, could explain these results.

In addition to mental health disturbances, many research studies reported that, due to homeschooling and remote working during this time, feelings of fatigue, headache, and discomfort increased among university students. All of these situations resulted in low student motivation and task engagement, and low acceptance of online learning during the pandemic, which could have directly affected their academic performance [3,5,17,18,19].

There is evidence that COVID-19 was a tremendous hindrance to the learning process of university students, with anxiety, stress, and depression conditions hampering online learning [20]. However, there is no conclusive agreement in the literature regarding the students’ academic performance in virtual classes throughout the COVID-19 pandemic.

In this regard, different findings [21,22], including those reported by Meiyi and Liu [23], indicated that COVID-19 negatively affected students’ performance and was positively correlated with anxiety. In contrast, other researchers revealed that improvements in academic performance were achieved by students during the pandemic as compared to pre-pandemic students [24,25].

Most of the previously described studies were conducted in Asia and Europe, where the disease first spread. However, to the best of the author’s knowledge, no studies have been conducted on the psychological health and/or academic performance of the Colombian university student population during the COVID-19 lockdown.

Although these lockdown periods have ended, this information is still important to university institutions. This knowledge could lead to the implementation of policies tracking and monitoring affected students during future pandemics.

Thus, considering the available evidence, we hypothesized that there is a positive association exists between cases of depression and anxiety and low academic performance. Furthermore, we expect females to be more affected than males by the studied variables.

The purposes of this study were to: (I) describe the frequency of low academic performance according to sociodemographic characteristics, (II) describe the frequency of depression and anxiety, and (III) assess the association between depression and anxiety with academic performance, stratified by sex, during the COVID-19 pandemic, in students from a public university located in Bucaramanga, Colombia.

## 2. Materials and Methods

### 2.1. Study Design

An analytical cross-sectional study from a public institution in Colombia was carried out from September to December 2020. The country was already in economic reopening but mobility restrictions were active in most of the main cities.

### 2.2. Procedure

Active students from the technological and professional levels of the socioeconomic faculty (physical activity and sports sciences and socio-economic sciences programs) of the Unidades Tecnológicas de Santander were recruited. This institution is one of the largest public institutions of higher education in the country, and is located in Bucaramanga, Colombia. An invitation was sent through administration channels (institutional email and Team’s groups) between 3 September 2020, and 21 September 2020, and was accessible via an anonymous link. In addition, the Dean of the faculty sent a reminder through the institutional web.

To take part in this study, participants had to meet the following inclusion criteria: (i) being enrolled at the university; (ii) being between 16 and 59 years of age. Exclusion criteria: (i) students with severe physical, psychological, or cognitive impairments that make it difficult to answer the questionnaires. Each participant signed an electronic informed consent form, along with an informed consent from their tutor, where the evaluation procedure was explained in a digital format.

The data collection was performed with a questionnaire via Microsoft Forms^®^, which was prepared based on the same questions as the original questionnaire. It was digitized for its online application. Survey data collection stopped on 22 September 2020, and these data were downloaded. A self-administered survey that took approximately 20 min to complete was offered to the students. It included 4 components: (1) sociodemographic characteristics, (2) perception of academic performance, diagnoses of COVID-19 in students and family members, (3) The Hospital Anxiety and Depression Scale (HADS) as a mental health assessment tool, and (4) grade points [23]. In the subsequent section, a thorough explanation of each variable is provided. The data were collected in a Microsoft Excel^®^ spreadsheet. Any participant with missing data for one or more of the items of interest (psychological mental health, age, grade point average, etc.) in the survey was removed from the dataset.

This research, its risks, and the privacy of the collected data are defined as “risk-free” research according to Resolution 8430/1993 of the Ministry of Health and Social Protection of Colombia [24]. The identification of each participant was coded and maintained under the principles of the Declaration of Helsinki, and the protocol was approved by the Ethics Committee for Human Beings from the Unidades Tecnológicas de Santander, no. 0010-2020/02.06.2020.

### 2.3. Mental Health Variables

The mental health variables were assessed with the Hospital Anxiety and Depression Scale Questionnaire (HADS), which was developed in 1983 by Zigmond and Snaith [26]. This is a self-applied instrument composed of 14 items that consider cognitive and affective dimensions. The evaluation of the psychometric properties of this instrument showed a high internal consistency (Cronbach’s alpha: 0.83–0.85), and high test–retest reliability (r = 0.75) [27]. The response options are Likert-type, ranging from zero to three (minimum score of 0 and maximum score of 21 for each subscale). A score from 0 to 7 indicated the absence of clinically relevant anxiety and/or depression, a score from 8 to 10 indicated doubtful cases, and from 11 to 21 indicated the presence of relevant symptoms with definitive cases [26]. The questionnaire has been translated and validated into Spanish, and utilized in previous studies conducted by different authors [27,28,29].

### 2.4. Grade Point Averages

The academic grades of the previous and current semesters were requested from each student. The students attached a screenshot of their academic records to verify that the information was true. According to the weighted average of the academic semester, the grades were determined. We evaluated the distribution of the academic grades, and we determined that a low academic performance was a score below the 5th percentile of the academic grades in the sample of the study.

### 2.5. Covariables

Covariables include sociodemographic characteristics, the perception of academic performance, and diagnoses of COVID-19 in students and family members. Students provided information about sex, age, marital and socioeconomic status, the program in which he/she was enrolled, time to study and grade. Moreover, students provided information on how the COVID-19 pandemic affected their academic performance (negative/positive), whether they were diagnosed with COVID-19 (yes or no), and whether their relatives were diagnosed with COVID-19 (yes or no).

### 2.6. Statistical Analysis

Firstly, descriptive analyses were performed to characterize the sample and determine the prevalence of low academic performance and the frequency of depression and anxiety categories in the total sample. All the analyses were stratified by sex.

Secondly, the sociodemographic characteristics, perception of academic performance, and diagnosis of COVID-19, according to low academic performance (Yes/No), were compared.

A Poisson regression with a robust error variance was used to assess the association between the depression and anxiety categories with the prevalence of low academic performance in the univariate and multivariable analyses stratified by sex. In multivariate analyses, the variables were selected using a backward stepwise selection strategy, and the variables with *p*-values less than 0.20 were maintained in the final model [30]. Sociodemographic characteristics (age, socioeconomic status, marital status), academic program, time to study, grade, self-perception of how the COVID-19 pandemic affected academic performance, whether students were diagnosed with COVID-19 and whether relatives were diagnosed with COVID-19 were included in the multivariate analyses. Considering the number of predictors, a backward stepwise selection was performed because it leaves only the most important variables in the model [31]. Prevalence ratios (PR) and 95% confidence intervals (95% CI) were calculated. Statistical significance was defined as a two-tailed *p*-value below 0.05. All analyses were performed using STATA^®^ version 12.0.

## 3. Results

### 3.1. Characteristics of the Participants

The potential participants were 2143. However, after excluding 185 datasets because of missing values for psychological mental health, age, or sex (Figure 1), a total of 1957 participants (males; *n* = 867 and females; *n* = 1090) was included in the present study.

The mean age of the students was 22.3 ± 5.4 years. The results showed that 55.7% were women, 93.4% were students from low socioeconomic backgrounds, 87.9% were single, 59.8% belonged to a program related to social and economic sciences, and 43.05% were in the second year of their career. With respect to COVID-19, 3.6% reported having been diagnosed with this illness, and 12.5% had relatives with a diagnosis for this virus. Finally, 51.5% perceived that the COVID-19 pandemic negatively affected academic performance; however, 53.1% rated their academic performance in the last semester as “good” (Table 1).

### 3.2. Mental Health Findings

Figure 2 shows the findings on anxiety and depression in the total sample and stratified by sex. The key findings were the frequency of students classified as “definite cases” of anxiety (2.5%), and depression (8.2%) in the total sample. In the stratified analysis, a higher prevalence of anxiety and depression was identified in women (3.8% and 10%, respectively), as compared with the prevalence of anxiety and depression in men.

### 3.3. Frequency of Low Academic Performance according to Sociodemographic Variables

Table 2 shows the frequency of low academic performance according to the sociodemographic characteristic. The main findings include a higher rate of low academic performance in men (4.7% *p* = 0.014), students enrolled in a sports science program (5.1% *p* = 0.003), students enrolled in the last year (10.5% *p* < 0.001), and students that perceived that before the pandemic, they did not have low academic performance (36.5% *p* < 0.001).

### 3.4. Association between Academic Performance and Mental Health Findings

Figure 3 shows the results of the unadjusted association between low academic performance and anxiety and depression, stratified by sex. The main findings include: (I) A lower proportion of women with a low academic performance classified as “definitive cases” of anxiety as compared with women without a low academic performance. However, this finding was not significant (3.8% vs. 0%, *p* = 0.20); (II) A lower proportion of women with a low academic performance, classified as “definitive cases” of depression, as compared with women without a low academic performance (6.9% vs. 10.1%, *p* < 0.05); and (III) A higher proportion of men with a low academic performance, classified as “definitive cases”, as compared with men without a low academic performance (12.2% vs. 5.4%, *p* < 0.05).

Figure 4 shows the results of the multivariable analysis performed with the Poisson regression. The findings are described below. Among the anxiety categories, definitive cases were not associated with a low academic performance (PR = 0.0002) in both women and men. However, women classified as “doubtful cases” of anxiety were associated with low academic performance (PR = 2.0 95% CI: 1.10–5.18, *p* = 0.026). Among depression categories, definitive cases and doubtful cases were not associated with low academic performance.

## 4. Discussion

This study aimed to assess the association between depression and anxiety and academic performance in university students who were enrolled in programs related to the social and economic sciences during the COVID-19 pandemic. The findings of this study confirm previous research, showing a low academic performance and disturbances in mental health due to the lockdown in this population [9,17,18]. Furthermore, a positive association between the severity of depression and anxiety and lower grades was found in university students, especially women.

### 4.1. Academic Performance and Lockdown

The results of the present study show that more than half of the population (51.5%) reported that the COVID-19 lockdown had a negative influence on their academic performance (Table 1). There are diverse possible explanations for this. Firstly, Colombia was ranked as one of the worst countries (#62) in terms of connectivity, speed, quality, affordability, infrastructure, and security in 2020 [32]. Thus, learning and work efficiency could have decreased among university students in our research during the lockdown, possibly due to limitations in the acquisition of the necessary technology and/or a lack of adequate digital connections.

Secondly, before the pandemic, fewer than 10% of university students received virtual classes in Colombia [33]. In this sense, the participants in our study may have had to deal with an unfamiliar scenario and restricted access to digital tools to cope with their studies, which could have had a negative impact on learning and academic performance.

Similarly, previous findings showed that, in undeveloped countries, online classes were a barrier to the students’ learning process. In this regard, a study showed that poor academic performance was due to students not being able to afford Internet services [34]. Also, researchers discovered that, during the lockdown, students were confronted with other difficulties, such as troubles in the response time, a lack of traditional classroom socialization, and insufficient face-to-face interaction with the professor. All these situations made them struggle to focus during online teaching sessions, resulting in a bad academic performance.

Likewise, Haider et al. [35] studied the psychosomatic impact of digital tools for online classes during the lockdown, indicating that 49.9% of the students pointed out that virtual classes were responsible for their low academic performance. Moreover, 55.5% of the participants mentioned that the volume of assignments via e-learning led to confusion, frustration, and lower grades.

Other researchers also found a significant negative impact of the COVID-19 lockdown and online education on academic performance [36,37,38,39]. The perception of conventional teaching as more effective and less fraudulent than online teaching was reported by Hanafy et al. [40]. This was confirmed in different studies on Romania and Saudi university students, which reported online education as being associated with being less motivated [41,42].

Nevertheless, contrary to our findings, the study by Gonzalez et al. [24] reported that the lockdown had a positive effect on learning strategies and study routines in the sample. Similarly, a study with 30,383 university students revealed a beneficial effect on their academic performance during the worldwide lockdown in countries located in Oceania, North America, and Europe [43]. This could be because the transition was fast in these developed countries and, during this process, the teaching staff offered sufficient support to the students. In addition, countries belonging to these continents were better placed than Colombia in the Digital Quality of Life Index 2020 ranking [32].

In our research, we found that sports science students showed a lower academic performance (5.1% vs. 2.6%; *p* = 0.003) than social and economic sciences students. A possible explanation for this could come from the curriculum of the sport sciences degree, which has historically lacked a focus on digital skills, concentrating courses on developing the practical abilities necessary for fieldwork. In this regard, the study by Liang et al. [44] demonstrated that only 11.7% of the evaluated physical education teachers had minimal digital knowledge. Furthermore, Swim et al. [45] reported that students demonstrated an absence of essential skills in Information and Communication Technologies, reporting the necessity to improve these competencies.

The impact of the COVID-19 lockdown on this variable was higher among students who were in their last year of their careers (Table 2). One possible reason for this could be that, in this academic year, they have to obtain some practical skills that cannot be developed in online classes.

### 4.2. Mental Health Psychological and Lockdown

The restrictive strategies imposed by governments worldwide, such as the reduction in social contact, home education, and remote work, were helpful in controlling the outbreak of the virus. Nevertheless, they caused psychological illnesses in different populations, which deserves additional study [46]. Therefore, another objective of this study was to investigate the impact of confinement on the mental health of university students, focusing on differences in the sex category. For this purpose, we used the HADS score as an overall measurement of the participants’ mental health.

One of the main concerns for several years has been university students’ mental health. A survey of 18,875 participants before the COVID-19 pandemic showed that 37% of university students had experienced an episode of depression, with 8% declared to have had suicidal thoughts in the past year [47]. In addition to lower grades, bad relations with colleagues, and lower levels of engagement on campus have been considered as sources of the alteration in mental health [48].

According to a scientific brief released by the WHO, in the first year of the COVID-19 pandemic, the global prevalence of anxiety and depression increased by 25%. Different factors such as social isolation, reduced ability to work, lack of support from loved ones, and lack of engagement in their communities could have impacted mental health [49]. In the literature, several studies have reported that the COVID-19 pandemic intensified this problem in the university population experiencing mental illness during the lockdown [50,51,52].

The prevalence rates of anxiety and depression in our study were 2.6% and 8.1% (Figure 2A), respectively, with these being higher in women, and extremely lower than in other studies. These results could be explained, in part, by the “Latin American phenomenon”. This theory proposes that the quantity and quality of social relationships and family life of citizens in this part of the world lead them to have higher than average levels of well-being and happiness, despite high levels of inequality [53].

In a US survey, students who reported a lower percentage of stress and anxiety during the pandemic, as in our results, attributed this to the time saved due to reduced schoolwork, pursuit of hobbies, or not being obliged to interact with others [13]. Additionally, because of the compulsory confinement, the cessation of academic work, the minimization of social interaction, and living with parents [6], students indicated feeling more depressed as compared to three years ago, according to Sazaki et al. [54].

In contrast, several studies [13,55,56] reported an increase in students’ levels of stress/anxiety during the pandemic and confinement period. In a survey of 8004 French students, 43.0% suffered from depression and 39.1% from anxiety [57]. Similar results were obtained from 2031 university students in the USA, where 71.26% of the participants stated that their stress/anxiety levels increased during the pandemic. Odriozola et al. [55] showed that moderate to extremely severe anxiety and depression symptoms were observed in 21.34% and 34.19% of Spanish students. The very high prevalence reported in these studies could be due to the higher percentage of COVID-19 infections in the evaluated sample. Along with this, distress about contracting the virus was also found to negatively impact students’ mental health, as previously reported [51,58,59]. In addition, many studies have shown that economic stressors such as food insecurity, low economic status, and a lack of social support from the government have been observed to affect mental health in the university population [51,58,60].

The results of the present study showed that women with “definitive cases” of anxiety and “doubtful cases” of depression were associated with a low academic performance in a higher proportion than men (Prevalence Ratio = 2.0 95% CI: 1.10–5.18, *p* = 0.026). In this regard, sex has been identified as one of the main risk factors that influenced the mental health of university students during the pandemic, with females being more affected than males [16,56,61,62].

In our study, sex was also considered as a determinant variable because of the association between depression and low academic performance (Figure 3A,B). Building on that point, a growing body of evidence supports the idea that being a woman is considered a notable risk for aggravating depression symptoms, owing to endocrinological differences determined by reproductive hormones [63,64,65]. During the pandemic, this prevalence increased due to stressors related to reproductive functioning and factors such as intimate partner violence, pregnancy and fertility issues [64].

In addition, different socioeconomic factors, including the inequality in access to work opportunities and income, could explain these results. In Colombia, females significantly receive lower wages than their male counterparts, and had more labour difficulties during the pandemic [66]. A report by the International Labour Organization (ILO) established that the national female unemployment rate in Colombia rose from 13.7% between July and September 2019 to 22.8% in the same period in 2020, while the unemployment rate for men rose from 8.3% to 13.9% in the same period in 2020 [65].

Recent evidence provides support the idea that mental health problems had a negative impact on academic work, causing multiple absences and disturbances in the prefrontal cortex and hippocampus’s functionality [67,68,69]. All of these alterations can be expressed in erratic cognitive functions, memory loss, and low retention, making sufferers more likely to perceive themselves as less academically competent [67]. Our findings are similar to those reported by Barbosa-Camacho et al. [70], where 610 Mexican students from public and private universities were surveyed in 2020 during the lockdown. This study reported that female students had a greater risk of depression and anxiety as compared to males, and significant effect was shown regarding the severity of mental health problems in the Academic Self-Concept Scale (ASC). In this sense, female students with moderate and severe mental health problems had lower ASC scores than those with no or minimal conditions.

To the best of our knowledge, this study is the first in Colombia to investigate the association between anxiety and depression during the COVID-19 pandemic and academic performance. The e-questionnaire allowed for us to assess different variables in university students while maintaining the WHO-recommended “social distance” during the COVID-19 pandemic, which would otherwise not have been possible. Additionally, the use of previously validated assessment tools is another strength of the study.

## 5. Limitations

The current study has several limitations. First, we used an online self-report survey in this study to assess anxiety and depression symptoms, which may have been subjected to social desirability and memory recall bias. Secondly, the cross-sectional design and a lack of a baseline may preclude the detection of possible bias in the measurements of the different variables. Although the number of respondents is large, it represents only 7.3% of enrolled students, which limits the generalization of the results.

Another limitation is that the tools designed specifically for the COVID-19 pandemic, such as the coronavirus anxiety scale (CAS), were not used. Finally, our research included students from a publicly owned university of low socioeconomic status, and did not consider previous mental health problems, experiences, or other psychosocial conditions and medical prescriptions, which might have been present and affected the students’ mental health. Finally, future research should include participants from all socioeconomic statuses, and explore the cause–effect between the severity of anxiety and depression with poor academic performance. Moreover, the way in which we determined academic performance in our study did not allow for us to compare our results with other research.

The results from this study must be considered in the context of the first wave of the COVID-19 pandemic, while people were quarantined.

## 6. Practical Applications

The study’s main findings provide evidence of implications for the mental health of university students and their academic performance. They also identify risk groups, such as female students with low academic performance. As a result, it is important to create programs in universities that concentrate on the creation of strategies to improve academic performance as well as the management of anxiety and depression.

Therefore, university professors and directives can expand the selection of locations that encourage social interaction and the practice of sports or other activities that support a healthy lifestyle. This is crucial to easing the transition to university life, which, in this instance, is made more difficult by the social, cultural, and national context.

## 7. Conclusions

The COVID-19 pandemic had a strong impact on the traditional teaching/learning methods of academic institutions around the world. Even though online learning was a helpful and safe tool during the COVID-19 pandemic, it was not as effective as conventional learning. In the present study, we found a positive association between anxiety, depression, and low academic performance, with a predominance in women. Furthermore, this is the first study on the impact of the COVID-19 lockdown on the mental health and academic performance of Colombian university students. These data could also be used to propose public strategies to reduce the risk of a negative impact on mental health and academic performance in the university student population, paying special attention to females.

## Figures and Tables

**Figure 1 healthcare-11-02024-f001:**
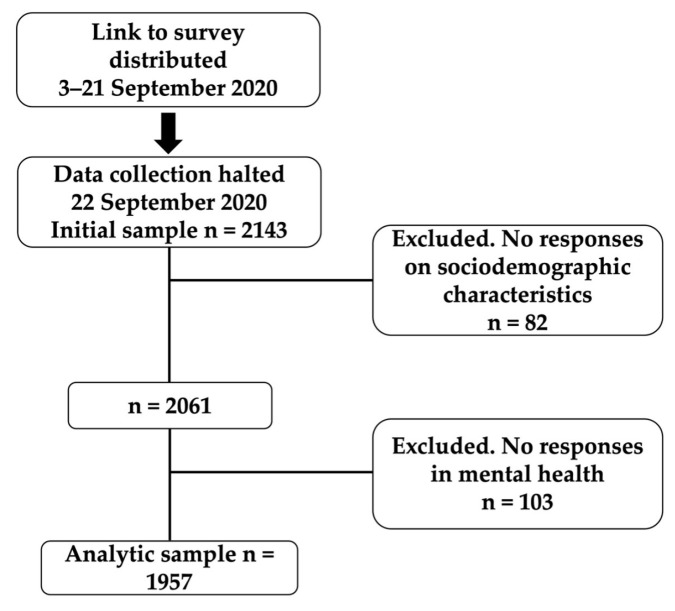
Participants flow to achieve the sample size (*n* = 1957).

**Figure 2 healthcare-11-02024-f002:**
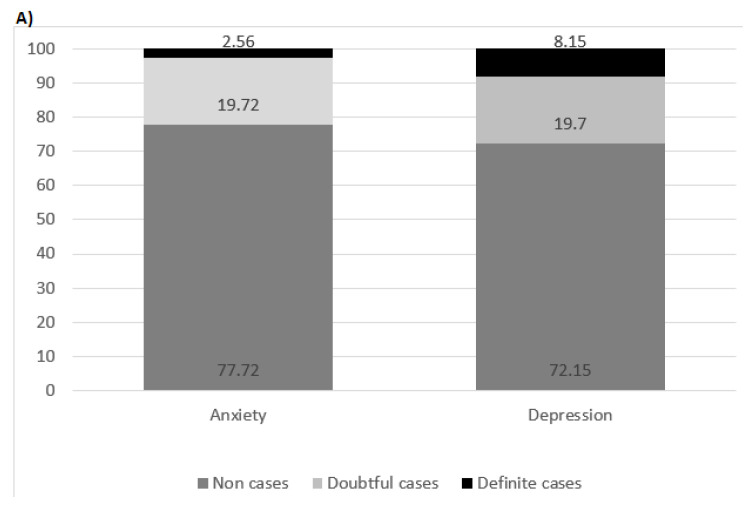
(**A**) Categories of anxiety and depression according to the total sample shown as percentages. (**B**) Categories of anxiety and depression stratified by sex shown as percentages.

**Figure 3 healthcare-11-02024-f003:**
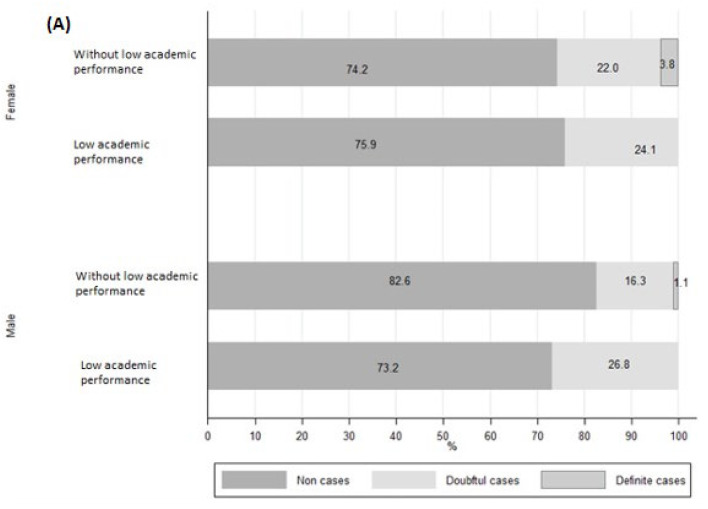
(**A**) Categories of anxiety of students with and without low academic performance, stratified by sex, shown as percentages. (**B**) Categories of depression of students with and without low academic performance, stratified by sex, shown as percentages.

**Figure 4 healthcare-11-02024-f004:**
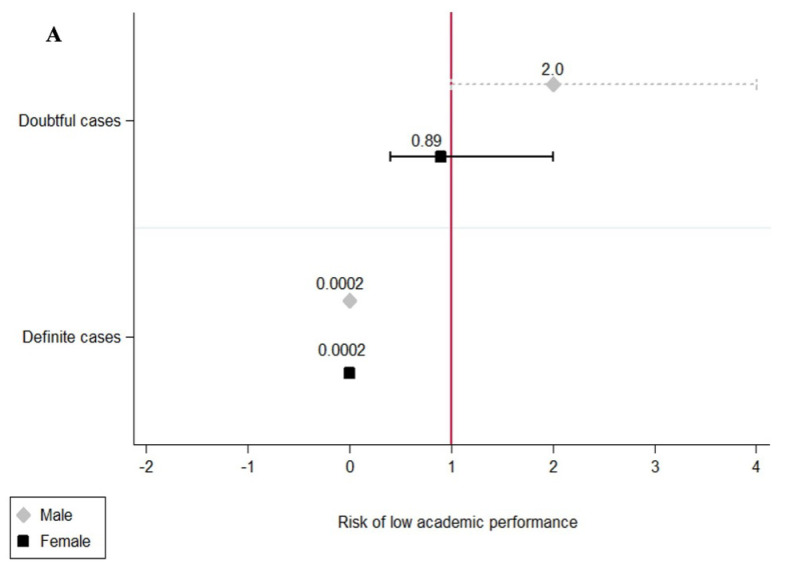
(**A**) Prevalence ratio of low academic performance according to anxiety categories stratified by sex, (**B**) Prevalence ratio of low academic according to depression categories stratified by sex.

**Table 1 healthcare-11-02024-t001:** Sociodemographic characteristics of the sample (*n* = 1957).

Characteristics	*n* (%)
Sex	
Male	867 (44.3)
Female	1090 (55.7)
Socioeconomic status	
Low	2006 (93.4)
Middle	129 (6.0)
High	12 (0.6)
Marital status	
Married	79 (4.0)
Cohabiting	150 (7.7)
Divorced	8 (0.4)
Single	1720 (87.9)
Program	
Sports Science	787 (40.2)
Social and Economic Sciences	1170 (59.8)
Time to Study	
Daytime	1100 (59.2)
Nighttime	757 (40.8)
Grade	
1	216 (11.5)
2	809 (43.0)
3	464 (24.7)
4	255 (13.6)
5	137 (7.3)
How did the COVID-19 pandemic affect academic performance?	
Neutral	949 (48.5)
Negative	1008 (51.5)
How would you rate your academic performance in the last semester?	
Excellent	250 (12.8)
Good	1040 (53.1)
Fair	588 (30.1)
Poor	79 (4.0)
Low academic performance in the last semester—before COVID-19	
Yes	151 (8.4)
No	1639 (91.6)
Have you ever been diagnosed as having COVID-19?	
Yes	70 (3.6)
No	1887 (96.4)
Have your relatives ever been diagnosed as having COVID-19?	
Yes	244 (12.5)
No	1713 (87.5)

**Table 2 healthcare-11-02024-t002:** Sociodemographic characteristics of university students with and without a low academic performance (*n* = 1957).

	Low Academic Performance		
Characteristics	No	Yes	
	*n* (%)	*n* (%)	*p* *
Sex			0.014
Male	826 (95.3)	41 (4.7)	
Female	1061 (97.3)	29 (2.7)	
Socioeconomic status			0.583
High	10 (90.9)	1 (9.1)	
Middle	118 (95.9)	5 (4.1)	
Low	1759 (96.5)	64 (3.5)	
Marital Status			0.942
Married	221 (96.5)	8 (3.5)	
Single	1666 (96.4)	62 (3.4)	
Program			0.003
Sports Science	747 (94.9)	40 (5.1)	
Social and Economic Sciences	1140 (97.4)	30 (2.6)	
Time to Study			0.200
Day time	1054 (95.8)	46 (4.2)	
Night time	734 (96.6)	23 (3.4)	
Grade			<0.001
1	136 (99.3)	1 (0.7)	
2	244 (95.7)	11 (4.3)	
3	446 (96.1)	18 (3.9)	
4	797 (98.5)	12 (1.5)	
5	194 (89.8)	22 (10.2)	
How did the COVID-19 pandemic affect academic performance?			0.228
Neutral	920 (96.9)	29 (3.1)	
Negative	967 (95.9)	41 (4.1)	
How would you rate your academic performance in the last semester?			0.116
Excellent	246 (98.4)	4 (1.6)	
Good	1005 (96.6)	35 (3.4)	
Fair	562 (95.6)	26 (4.4)	
Poor	74 (93.7)	5 (6.3)	
Low academic performance in the last semester—before COVID-19			<0.001
No	96 (63.6)	55 (36.4)	
Yes	1629 (99.4)	10 (0.6)	
Have you ever been diagnosed as having COVID-19?			0.102
No	1822 (96.6)	65 (3.4)	
Yes	65 (92.9)	5 (7.1)	
Have your relatives ever been diagnosed as having COVID-19?			0.524
No	1650 (96.3)	63 (3.7)	
Yes	237 (97.1)	7 (2.9)	

* Chi-Square.

## Data Availability

The dataset presented in this study is available from the corresponding author on reasonable request.

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
