# Peer review of "Cross-Sectional Analysis of Colombian University Students’ Perceptions of Mental Health during the COVID-19 Pandemic: Repercussions on Academic Achievement"

_healthcare, 2023, doi:10.3390/healthcare11142024_

Round 1
Reviewer 1 Report
Abstract - Abstract need to highlight the importance or gaps to attract readers. Tools should be mentioned first before number of students participated A 20 total of 1957 students with median age of 22.2 years participated in the study. Need to summarize findings on prevalence To close abstract with implication of findings Introduction Literature has been summarized with gaps highlighted for Colombian population / Need to justify the importance despite lockdown period has ended Methodology 2.1 Although study site is anonymous, need to give context to readers - public/private, which program that was involved, how was lockdown changed the teaching and learning for these program 2.2 Introduce inclusion and exclusion criteria before data collection pocedures 2.4 - not clear by review by each student (self declare or verified with institution database?) Question of how covid affect academic performance and rate academic performance are quite subjective across participants - How to ensure objectivity in responses? Results Figure 1 can be removed, already described in text Figure 2 and 3 suggest to be transformed in tables for better readability The rationale of analysis in Table 2 is not related to the study objective Need to also state the insignificant findings for Poisson as this is the main objective Discussion Need to cite - As expected, the findings of this study 233 are consistent with several previous studies Paragraph 2 is repetition of result - please discuss findings instead In our research, we found that sports science students showed lower academic per- 291 formance (5.1% vs. 2.6%; p=0.003) than social and economic sciences. - kindly discuss or postulate explanation 4.2 third para should be in Introduction instead of Discussion General Manuscript requires proofreading to improve the scientific writing and sentence structure.Author Response
We thank reviewer 1 for his/her critical evaluation and valuable comments. We have considered his/her recommendations and suggestions. Itemized responses are listed below. All the modifications have been marked in red throughout the manuscript as the editor recommended. Please see the attachment.

Reviewer 2 Report
Thank you very much for give me the opportunity to review this interesting paper entitled Cross-Sectional Analysis of Colombian University Students’ Perceptions of Mental Health during the COVID-19 Pandemic: Repercussions on Academic Achievement. Altogether it is an interesting and quite well written study that aimed to investigate the association of depression and anxiety on university students’ academic performance during the COVID-19 lockdown conducting a cross-sectional study, through an online survey during the lockdown. As authors noted most of the studies at his level have been conducted in Asia and Europe, but not in Colombia.
The objective is well defined, and citations are current.
The procedure and data analyses process are amply described, the sample size is high, and the statistical analysis are coherent.
Say this, here are a few comments:
1. I'm more used to the expression procedure than to procedures in 2.2.
2. In a similar vein, I am not very sure of you organization in subheadings. For example, 2.1. Study design and sample, I am more used to the expression participants, and to the fact that even minimally some data will be offered that describes them in that section (although I understand that this broader description is made in the results section), and perhaps to that the characteristics of the study and its nature (or design) will be described in the procedure section.
3. I am also more used to the fact that when the p is significant it is written with italics (p)
4. Please check the format (p. e. line 53, line 54 “COVID-19 lockdown in 2031”, line 155 figure 1 in bold, it seems that line 218 after the title of the figures is missing a space, and line 219 should start below)
5. Maybe the conclusion section might be improved, in example, adding some more arguments regarding the practical implications and suggestions for future research of your work.
In sum, it is an interesting study, in which the procedure and the research process are amply described.
Author Response
We thank reviewer 2 for his/her critical evaluation and valuable comments. We have considered his/her recommendations and suggestions. Itemized responses are listed below. All the modifications have been marked in red throughout the manuscript as the editor recommended. Please see the attachment.

Reviewer 3 Report
Dear Authors, I have compiled some comments and suggestions regarding the article after carefully reviewing your manuscript.
Here you will list the comments and suggestions:
The abstract contains important information and results from the study, but there are some areas where clarity and precision can be improved. Here are some suggestions:
- Specify the location of the study: If the study is limited to a specific university or region in Colombia, it is essential to specify that in the abstract.
- Clarity in sentence structure and results reporting: Some sentences can be difficult to understand due to complex phrasing or the order of information presented. For example, the sentence, "Women with negative low academic performance reported in more proportion 'definite cases' (3.8%) of anxiety compared with men (1.1%)." could be rephrased for clarity:.
- Consistent usage of terms: In the same vein, consider using consistent terms to describe groups. For instance, instead of "Women with negative low academic performance," use "Women with low academic performance."
- Clarify key terms: Definitions or brief explanations of key terms can aid understanding. For instance, the terms "definite cases" and "doubtful cases" could be clarified, as what they mean is not immediately apparent.
- Clarify the conclusion: While the study concludes that there's a positive association between the severity of anxiety and depression with poor academic performance, it might be helpful to specify if this means that increased severity of anxiety/depression correlates with lower academic performance.
- Add implications or recommendations from the study: The abstract would benefit from a brief mention of the study's implications or potential applications of the findings.
- Precision in statistical results: Make sure to include clear information about the statistical significance of your results. If possible, try to avoid words like "more," "less," "higher," etc. and instead provide the exact figures or percentages.
This introduction section provides a clear context, relevant background information and the study objective. However, a few improvements can be made to enhance clarity, specificity, and readability:
- Specify statistics: The first sentence talks about the number of cases and deaths worldwide. Adding a date or timeframe for these figures would be helpful.
- Improve sentence structures: Some sentences are long and complex, making them difficult to understand. These could be broken down into shorter, simpler sentences to improve readability.
- Improve flow: The paragraph starting with "Although several studies have shown..." seems abrupt. Consider integrating it more smoothly into the text.
- Citations: Some parts lack the proper referencing format, such as "[10](48.14%, n=960)". It would be better to write as "[10] reported that 48.14% (n=960) of the participants exhibited symptoms of depression."
- Consistency in terminology: Consistency in terms is important. For example, consistently use ‘university students’ or ‘college students’.
- The clarity in previous studies: When discussing the findings of previous studies, try to maintain consistency and clarity. For instance, the Italian study only mentions anxiety and headaches without mentioning depression. It may be better to consistently report on depression, anxiety, and other mental health indicators across all the studies.
- Clarify the problem statement: Consider explicitly stating the problem your research aims to address.
- Objective of the study: Consider providing more information about the expected outcomes of the study.
- Organization: The introduction seems disjointed in some areas, with some sentences not smoothly leading to the next. Consider reordering some parts for better flow.
- Data precision: When reporting the data from other studies, make sure that the values are not ambiguous. For instance, "A study from Italy of 3533 university students reported 13.1% anxiety and 43.6% for headaches during the lockdown" is a bit unclear. Do these percentages represent the proportion of students who reported anxiety and headaches?
The "Materials and Methods" section comprehensively outlines the study's design, procedures, data collection, and analysis methods. However, it can benefit from improvements in the following areas:
- Study design and sample details: When discussing the study's cross-sectional design and sample, be more specific about how the university students were selected, the criteria for their inclusion, and the reasons behind choosing this particular institution.
- Chronological order of information: The "Procedures" subsection mentions the inclusion criteria after detailing the recruitment and data collection process. Consider starting with the criteria for clarity.
- Clarity in data collection tools: Specify whether the Microsoft Forms® questionnaire was a pre-existing tool or designed specifically for this study. If it's a new tool, provide information about its validity and reliability.
- Inclusion criteria: It's unlikely that all students are "healthy (physically, psychologically, and cognitively)." Consider specifying criteria for what constitutes as 'healthy' in this context, or rephrase this criterion to something more precise like "without severe physical, psychological, or cognitive impairments."
- Explanation of methods: In the "Grade point averages" subsection, explain why you chose a score of <3.5 to indicate low academic performance. This would give more insight into your methods and reasoning.
- Definition of terms: The "Mental Health Variables" subsection refers to "clinically relevant anxiety and/or depression," "doubtful cases," and "definitive cases". Define these terms
- Missing Methodologies: The demographic and other variables used in the multivariable analysis aren't explicitly mentioned. Clarify which other factors were controlled for in the model.
- Statistical analysis: Explain why you chose a stepwise backward selection strategy in the multivariable analysis.
While the section Results is generally well written, several areas could be improved for clarity, readability, and precision.
- Introduction to Results: A brief overview or summary of the main findings could be beneficial in guiding the reader through the detailed results that follow.
- Grammar and Clarity: There are instances of awkward phrasing and unclear language. For example, the statement "Women’s prevalence was higher than those of men on anxiety (3.8%) and depression (10%)" could be reworded for clarity: "The prevalence of anxiety and depression was higher in women (3.8% and 10%, respectively) than in men".
- Descriptive Language: There are several instances where more descriptive language could be used to clarify the findings. Instead of "low socioeconomic status," consider "students from low socioeconomic backgrounds". Also, instead of "free union," perhaps use a more universally understood term such as "cohabiting" or "in a domestic partnership".
- Consistent Terminology: The terminology is not consistent, leading to confusion. For instance, in some parts, the paper refers to "low academic performance," while in others, it uses the term "negative low academic performance".
- Statistical Interpretation: Some of the statistics need a better explanation. For example, the statement "probability of low performance was 100% higher for women identified as “doubtful cases” of depression" could be clarified. Does this mean women with "doubtful cases" of depression were twice as likely to have low academic performance?
- Figure and Table Descriptions: All figures and tables should be clearly explained in the text, and a better description of the results should be presented.
The discussion section you provided appears comprehensive and follows the general conventions of a research paper, making its points clear with reference to related literature and studies. However, a few possible areas for improvement can be pointed out:
- Providing a Clear Summary: It is typical for the discussion to start with a brief summary of the main results found in the study. While the discussion does mention key findings, it could provide a more succinct summary of the key results at the beginning to help readers understand the context of the subsequent points.
- Enhancing Readability: The text has a high density of citations and references. While this is necessary in research papers, the information could be presented more easily. Try to summarize studies' findings in your own words, interspersed with key quotes or statistics, rather than relying heavily on direct citations.
- Improving Clarity: Certain sentences are too long or complex, making them difficult to understand. For example, the sentence at line 337 could be broken down into multiple simpler sentences.
- Discussion of Unexpected Results: It would be beneficial to discuss any unexpected results and explain why they might have occurred or what further research would be necessary to understand them. For instance, the low prevalence rates of anxiety and depression are noted, but there's no discussion or explanation of why these results might be significantly lower than in other studies.
- Including Recommendations for Future Research: While the discussion identifies some limitations of the study, it could also suggest specific directions for future research to address these limitations.
- Implications and Recommendations: The discussion could also include more about the potential implications of the study's results and recommendations for policy, practice, or further research. There is mention of this in the conclusion, but it could be expanded upon in the main body of the discussion.
- Contrasting findings: The comparison with studies that contradict your results needs to be more thorough. While you've included contradictory studies, a more explicit discussion about why your results may differ could be helpful.
-
Dear Authors, extensive editing of the English language is required; some comments and suggestions below:
- Grammar, punctuation, and stylistic errors: The language throughout the text should be revised for grammatical errors, clarity, and flow. While the grammar is largely correct, some sentences are unnecessarily complex or lengthy. This can make the paper harder to read and understand.
- Consistency in terms: Ensure that the same terms are used consistently throughout the paper. For instance, it's confusing to interchange terms like 'online classes', 'virtual classes', 'digital learning', etc. Decide on one term and stick with it throughout the paper.
- Avoid redundant phrases: For instance, "this is the first study in Colombia that investigated the association" could be rephrased as "this study is the first in Colombia to investigate the association".
- Direct and concise language: Some sentences can be simplified for ease of reading. For example, "we found that anxiety and depression had a positive association with low academic performance" can be rewritten as "we found a positive association between anxiety, depression, and low academic performance".
- Clarification of data: Some data points could be made clearer. For instance, "anxiety and depression in our study were 2.6% and 8.1%" – is this the rate of incidence among the study group? Also, the sentence "we found that sports science students showed lower academic performance (5.1% vs. 2.6%; p=0.003) than social and economic sciences" is unclear – it would help to clarify what the percentages represent.
- Avoiding anthropomorphism: For example, instead of saying, "the HADS score was used as an overall score to determine the participants’ state of mental health", it is better to say, "we used the HADS score as an overall measure of participants’ mental health".
- Phrasing of results and conclusions: The results and conclusions can be stated more assertively. Instead of "as expected, the findings of this study are consistent with several previous studies," you can say "the findings of this study confirm previous research showing that..."
- Specific terminology: Replace non-specific terms such as "poor academic performance" with more precise terms, such as "decreased grades", "increased course withdrawal", "delayed graduation", etc., depending on what "poor academic performance" specifically refers to in your study.
- Transitions: Improve the flow of information by using better transitional words or phrases. For example, instead of starting a paragraph with "Similarly", consider "Building on this point," or "Furthermore".
Author Response
We thank reviewer 3 for his/her critical evaluation and valuable comments. We have considered his/her recommendations and suggestions. Itemized responses are listed below. All the modifications have been marked in red throughout the manuscript as the editor recommended. Please see the attachment.

Round 2
Reviewer 3 Report
I want to thank the authors for incorporating the suggested improvements into the article. The revisions have certainly enhanced the clarity and depth of the study. However, it is worth noting that there are still some areas that could benefit from further improvement. These include providing more specific details and examples to support the arguments, addressing limitations in a more comprehensive manner, providing a clearer description of the research methodology, discussing potential implications in greater depth, offering a balanced perspective, and enhancing the overall clarity of the article.
Author Response
We thank reviewer 3 for their critical evaluation and valuable comments. We have considered their recommendations and suggestions. Itemized responses are listed below. All the modifications have been marked in blue throughout the manuscript as the editor recommended.
